# When Does Uncertainty Matter?: Understanding the Impact of Predictive Uncertainty in ML Assisted Decision Making

**Sean McGrath**                                              *sean_mcgrath@g.harvard.edu*
*Harvard University*

**Parth Mehta**                                              *mehta.parth117@gmail.com*
*Harvard University*

**Alexandra Zytek**                                              *zyteka@mit.edu*
*Massachusetts Institute of Technology*

**Isaac Lage**                                              *isaaclage@g.harvard.edu*
*Harvard University*

**Himabindu Lakkaraju**                                              *hlakkaraju@hbs.edu*
*Harvard University*

**Reviewed on OpenReview:** *https://openreview.net/forum?id=pbs22kJmEO*

## Abstract

As machine learning (ML) models are increasingly being employed to assist human decision makers, it becomes critical to provide these decision makers with relevant inputs which can help them decide if and how to incorporate model predictions into their decision making. For instance, communicating the uncertainty associated with model predictions could potentially be helpful in this regard. In this work, we carry out user studies (1,330 responses from 190 participants) to systematically assess how people with differing levels of expertise respond to different types of predictive uncertainty (i.e., posterior predictive distributions with different shapes and variances) in the context of ML assisted decision making for predicting apartment rental prices. We found that showing posterior predictive distributions led to smaller disagreements with the ML model's predictions, regardless of the shapes and variances of the posterior predictive distributions we considered, and that these effects may be sensitive to expertise in both ML and the domain. This suggests that posterior predictive distributions can potentially serve as useful decision aids which should be used with caution and take into account the type of distribution and the expertise of the human.

## 1 Introduction

As machine learning (ML) models are increasingly being deployed in critical domains such as healthcare and criminal justice, there has been a growing emphasis on the need for human interpretable models and predictions which can enable decision makers to decide if and how much to rely on these predictions. In order to interpret model predictions, the following two complementary approaches have been proposed in literature. The first paradigm involves building inherently simpler models such as decision trees/lists/sets (Letham et al., 2015; Rudin, 2019; Lakkaraju et al., 2016), point systems that can be memorized (Ustun & Rudin, 2016) or generalized additive models in which the impact of each feature on the model's prediction is explicitly given (Caruana et al., 2015). However, complex models such as deep neural networks and random forests seem to achieve higher accuracy compared to these inherently simpler models in several real world settings (Ribeiro

et al., 2016). Therefore, an alternate approach of constructing post hoc explanations to interpret these complex models has been proposed in literature (Ribeiro et al., 2016; Lundberg & Lee, 2017). Both simple models and post hoc explanations are attempts at creating more interpretable decisions aids.

In addition to the aforementioned simple models and explanations, other auxiliary information such as uncertainty associated with predictions could also potentially serve as useful decision aids. While there are reasons to believe that conveying information pertaining to predictive uncertainty might help decision makers in figuring out how to incorporate model predictions into their decision making, there is little empirical work that systematically explores the validity of this hypothesis (Bhatt et al., 2021). While some prior research has examined how predictive uncertainty in various contexts ranging from weather forecasting to public transit scheduling can be communicated to the public (Greis et al., 2016; Roulston et al., 2006; Fernandes et al., 2018), these works largely focus on the cases where uncertainty can be expressed using a low variance normal distribution. However, in reality, predictive uncertainties could be much more complex to reason about (e.g., posterior predictive distributions which are bimodal or normal with high variance). In such cases, interpreting predictive uncertainty and incorporating it in decision making is non-trivial, and is heavily driven by the end user's domain expertise and familiarity with ML.

To illustrate, let us consider the following example: if an ML model predicts that Alice should sell her car for $20,000, should she place an advertisement for that amount or rely on her background knowledge to price the car herself? If she knew the model was highly certain, the rational choice may be to defer to its prediction, but if it is highly uncertain, she may instead prefer her own. However this picture is complicated by both Alice's own level of expertise, and the form of the uncertainty associated with the prediction by the model. For example, what if Alice is an expert used car appraiser? Or what if the model is highly certain that the car will sell for either $17,000 or $23,000? A rational choice for Alice might look different between these cases, and it is unclear whether the human decision maker will even follow a rational strategy. Therefore, it is important to systematically study how different types of predictive uncertainty (e.g., posterior predictive distributions with different shapes and variances) and different contexts (namely decision makers with varying degrees of domain expertise and familiarity with ML) impact decision making.

In this paper, we explore how decision making is impacted when decision makers (end users) are shown estimates of predictive uncertainty. More specifically, we consider posterior predictive distributions as our measure of uncertainty, and study how posterior predictive distributions with various shapes and variances impact decision making in cases where decision makers have varying degrees of expertise in ML and the problem domain. We consider the setting where decision makers are only given limited information for the task at hand, and are shown the output of a ML model (which may consider features not available to the decision makers). While the goal is to understand how uncertainty affects the decision making process as a whole, in this work, we focus on systematically measuring how closely participants agree with the predictions of the model based on the factors described above. This allows us to separate the impact of uncertainty from the quality of the ML model. To the best of our knowledge, this work makes one of the first attempts at systematically exploring the aforementioned research questions.

To find answers to the above questions, we conduct a user study with 1,330 responses from 190 participants where each participant is asked to predict monthly rental prices of apartments in Cambridge, Massachusetts (MA). Participants were recruited from the platform Prolific as well as from research groups in various fields across multiple universities. We found that: 1) Showing posterior predictive distributions (regardless of their shapes and variances) led to smaller disagreements with the ML model's predictions compared to showing no uncertainty estimates at all. 2) Showing model predictions (whether or not posterior predictive distributions were also shown) helped equalized differences between people with and without domain expertise. 3) People with ML expertise agreed with the model most when provided with posterior predictive distributions that were normally distributed and had low variance. This suggests that posterior predictive distributions can potentially serve as useful decision aids which should be used with caution and take into account the type of distribution and the expertise of the human.

## 2   Related Work

**ML Assisted Decision Making**: Machine learning is increasingly being employed to assist human decision makers. For example, Kleinberg et al. (2017) examine ML assisted decisions in the context of the the criminal justice system, and Giannini et al. (2019) studies how ML can assist in a clinical care context. Consequently, there has been a lot of work in how decision makers make use of the output of ML models (Lai et al., 2021). In some cases, users may trust models even when the model's predictions are inaccurate (Poursabzi-Sangdeh et al., 2021). In other cases, users will ignore a model's prediction even when it is expected to perform better than humans (Yin et al., 2019). More generally, Skitka et al. (1999) found that people made more errors when aided by highly accurate automation, suggesting the existence of an automation bias that affects people's decision making with automated aids. We explore how showing uncertainty information for a specific prediction alongside an ML prediction can affect the downstream decisions made by a human decision maker.

**Model Interpretability**: Prior research has suggested that model interpretability can be extremely helpful in ML assisted decision making (Doshi-Velez & Kim, 2017). Different classes of interpretable models have been proposed (Wang & Rudin, 2015; Zeng et al., 2017; Letham et al., 2015; Bien & Tibshirani, 2009; Lakkaraju et al., 2016; Lou et al., 2012; Caruana et al., 2015) and studied in the context of human decision making (Lage et al., 2019; Poursabzi-Sangdeh et al., 2021). While these simpler interpretable models are often easier for users to interact with, complex models such as deep neural networks and random forests are often shown to achieve higher accuracy (Ribeiro et al., 2016). This motivated the development of post hoc explanation methods, including perturbation-based local explanation methods (Ribeiro et al., 2016; 2018; Lundberg & Lee, 2017; Slack et al., 2021), gradient-based local explanation methods (Selvaraju et al., 2017; Simonyan et al., 2014; Smilkov et al., 2017; Sundararajan et al., 2017), global explanation methods (Bastani et al., 2017; Lakkaraju et al., 2019), and other methods (Koh & Liang, 2017).

However, these post hoc techniques have been shown to have flaws. Rudin (2019) argued that post hoc explanations are not reliable, as these explanations are not necessarily faithful to the underlying models and present correlations rather than information about the original computation. There has also been recent work on exploring vulnerabilities of black box explanations (Adebayo et al., 2018; Slack et al., 2020; Lakkaraju & Bastani, 2020; Rudin, 2019; Dombrowski et al., 2019). Moreover, there is a growing literature on evaluating the effectiveness of post hoc explanations for ML assisted decision making (Doshi-Velez & Kim, 2017; Kaur et al., 2020; Bhatt et al., 2020; Hong et al., 2020; Lakkaraju & Bastani, 2020; Poursabzi-Sangdeh et al., 2021; Buçinca et al., 2020; Krishna et al., 2022), which has highlighted a number of challenges and subtleties. For instance, while prior studies have shown that decision makers can perform better when provided with ML predictions and corresponding explanations in various tasks such as medical diagnosis (Cai et al., 2019; Lundberg et al., 2018), data annotation (Schmidt & Biessmann, 2019), and deception detection (Lai & Tan, 2019), Bansal et al. (2021) argued that such improvements may be due to decision makers simply following the recommendations of highly accurate ML models rather than placing so-called appropriate reliance on the models (Lee & See, 2004). Indeed, Bansal et al. (2021) found that decision makers were more likely to follow the model's prediction when provided with explanations, regardless of the model's correctness.

We focus on studying the impact of predictive uncertainty on user behavior since it is often more easily attainable than a faithful and interpretable explanation. Based on insights from the explainable AI literature (Buçinca et al., 2020), we focus on the impact of predictive uncertainty on user behavior in an actual decision-making task rather on than proxy tasks or subjective measures such as user trust or preference.

**Predictive Uncertainty as Auxiliary Input**: Bhatt et al. (2021) outline key considerations for conveying uncertainty for ML assisted decision making. Many works have empirically studied how to best communicate uncertainty to decision makers in tasks ranging from reacting to weather forecasts to catching public transportation (Correll & Gleicher, 2014; Kay et al., 2016; Greis et al., 2016; Roulston et al., 2006; Fernandes et al., 2018; Kirschenbaum et al., 2014; Leffrang & Müller, 2021; Koval & Jansen, 2022). For example, Greis et al. (2016) finds that people earned more when shown uncertainty estimates in a farming game that depends on weather forecasts. They also found that the performance was best when showing a simpler uncertainty visualization than the full distribution. Roulston et al. (2006) finds that showing participants error bars in addition to a point estimate can help them make a larger profit in a game where they must decide when to

salt roads based on a snow forecast. They also find, however, that showing explicit probabilities does not lead to further improvements. Fernandes et al. (2018) explores the question of which types of uncertainty visualizations help users make the 'best' decisions according to a specified payoff function when catching a bus. They explore different uncertainty visualizations and find that all of them allow people to improve over time, and their preferred method – quantile dotplots – resulted in the most consistent performance. Kirschenbaum et al. (2014) studied the impact of spatial vs. tabular uncertainty representation on submarine detection and found that spatial uncertainty raised the performance of non-experts almost to that of experts. Motivated in part by a preprint of our work, Leffrang & Müller (2021) studied the impact of showing 95% confidence interval plots and ensemble displays in time series forecasts of the number of hospital beds occupied by COVID-19 patients. They found that users were less willing to follow the model's prediction when shown more salient visualizations of uncertainty, and called for further studies evaluating the impact of conveying uncertainty visualizations. Our work differs from previous research in that we systematically study how decision makers are impacted when shown posterior predictive distributions under several settings. We also consider expert decision makers – unlike most of these works – who may be able to work with more complex information about the probability distribution (Spiegelhalter et al., 2011).

Finally, some work has been conducted on how people make decisions with ML models that present uncertainty information (Zhou et al., 2017; Arshad et al., 2015). For instance, Zhou et al. (2017) conducted a user study to evaluate how uncertainty intervals around a point estimate affects users' predictive decision making. They found that showing users such uncertainty intervals increased trust, but only under conditions with low cognitive load. We study the impact of showing uncertainty information on downstream decisions, rather than relying on trust as a proxy for team performance. We also explore different features of the uncertainty distribution that may affect human decisions.

## 3 Goals and Study Design

### 3.1 Goals

In this project, we seek to understand how people make decisions when provided with the uncertainty of an ML model as an input to their decision process. The uncertainty of an ML model is auxiliary information provided by many ML models that can be used as an input to human decision making in addition to the model's prediction. Specifically, we explore how different factors related to the model's uncertainty distribution impact ML assisted decision making. We measure this through user agreement with the model's prediction before and after being shown the uncertainty information, which allows us to isolate the important factor of agreement with the machine prediction from whether the prediction is correct. We also study how ML assisted decision making differs depending on user expertise in the domain or in ML. We describe each of these factors in more detail below.

We operationalize the notion of predictive uncertainty in a Bayesian setting where predictions may be given as point estimates or as entire posterior predictive distributions. We show uncertainty information as a visualization of the model's full posterior predictive distribution, in addition to its prediction marked at the mean of the posterior predictive.

We study the effect of 2 factors related to the uncertainty distribution: the shape of the uncertainty distribution, and its variance. Different shapes of distributions have different properties that may both affect how much people are willing to take them into account when making decisions, and what the optimal way to take them into account may be. We study 3 shapes of posterior predictives: a normal distribution, a skewed distribution, and a bimodal distribution. A normal distribution is commonly studied in past work (e.g., Fernandes et al. (2018)), and has the property that its mean is equal to its mode. This is not true however for the bimodal distribution, which has 2 modes and very little probability near its mean, or the skewed distribution which has a mode to one side of the mean and a long tail of reasonably high probability outcomes. Whether people will still update their estimates to reflect the ML model's prediction when presented with these distributions with more complicated properties is an open question.

We also explore the effect of variance on people's behavior. We wish to answer the question of how much uncertainty affects people's decisions when the variance of the posterior distribution is larger vs. when it is

smaller. Are people more likely to agree with an ML prediction when the uncertainty around the prediction is low? That is, do they correctly interpret and respond to uncertainty? And how does this compare to when only shown a point estimate? We study this in the context of the normal distribution.

Moreover, we explore how decision differs between experts (in either the domain or in ML) and non-experts. Will people with differing levels of ML expertise feel more or less compelled to follow the machine's prediction regardless of the provided uncertainty? Will they respond more or less strongly to the presented uncertainty estimates? Orthogonally, will people who know more about the domain we study feel more confident in their prediction and therefore place comparatively less weight on the machine's prediction? We study the relationship between these notions of expertise and decision making with various forms of uncertainty.

Finally, we study the correlation between subjective measures of trust and perceived usefulness of the ML model and participant agreement with the model. In summary, our main research questions are:

- Does showing predictive uncertainty affect how closely people follow model predictions?

- Does the effect of showing predictive uncertainty depend on the type of uncertainty – either the shape or the variance of the distribution?

- Do participant with more expertise, either in the domain or in ML, more closely follow model predictions?

### 3.2 Study Design

In this section, we describe the details of the user study that we carried out to answer the research questions outlined in the previous section.

#### 3.2.1 Domain and Model

Our user study was based on predicting the monthly rental prices of apartments in Cambridge, MA. Participants were provided with the number of square feet, number of bedrooms, and the relative size of each apartment (i.e., whether the apartment size is smaller than average, average, or larger than average for its number of bedrooms). We provided only these features to the participants to align with typical real world scenarios where decision makers only have partial information available.

To set realistic apartment prices, we collected details such as the monthly rental listing price, square footage, and number of bedrooms of 75 apartments in Cambridge, MA from Zillow.com in November 2019. A linear regression model was fit to this data to estimate the monthly rental listing price using the square footage and number of bedrooms as features. This model was then used to set the point estimates for each of the apartments in the study.

The posteriors were not generated from the model, but were instead hand generated to embody the properties we aim to study. These are described in more detail below.

All participants in the study were shown the same 10 hypothetical apartments. Specifically, the number of bedrooms and square feet for the 10 hypothetical apartments were respectively (1, 500), (1, 800), (2, 500), (2, 800), (2, 1100), (3, 800), (3, 1100), (3, 1400), (4, 1400), and (4, 1700). Three of these (namely, (2, 500), (3, 1100), (4, 1700)) were used as practice examples for participants to familiarize themselves with the study (see Section 3.2.3 for more details).

#### 3.2.2 Conditions

We consider 5 conditions in our study each of which corresponds to different properties of the posterior predictive distribution shown to the participant:

**No Uncertainty**: Participants were only given access to the model's point estimate of the predicted rental price for each apartment.

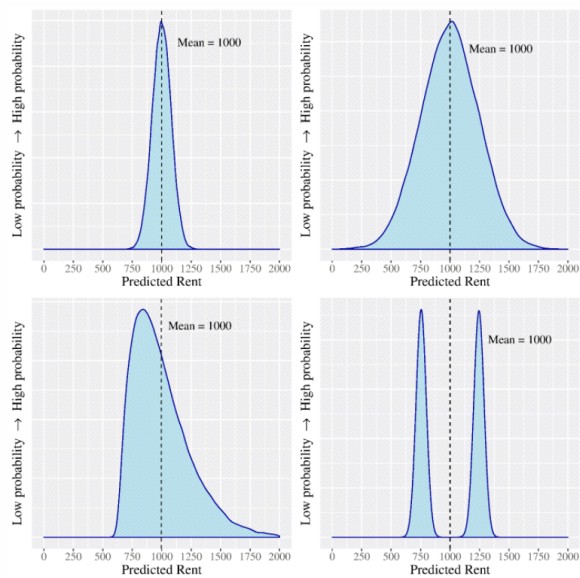

In addition to its estimate of the price, the machine learning model may also tell you how uncertain it is about its estimate. If it provides this information, you will see a graph with apartment prices on the x-axis, and the probability the machine learning model places on the price in the y-axis.

The higher the blue line is for a certain price, the more confident the machine learning model is that that's what the apartment costs. The dotted black line represents the machine learning model's average estimate of the apartment price.

Uncertainty can look different in different circumstances. The 4 examples below correspond to qualitatively different types of uncertainty that the machine learning model may have.

Figure 1: The uncertainty tutorial page given to participants before completing the task. The tutorial shows 4 examples of uncertainty distributions that correspond to the 4 types of uncertainty shown in different conditions. This same page is shown regardless of the condition the participant is randomized to. The text tells participants how to interpret the $x$- and $y$-axis. The dotted black line marks the mean of each distribution.

**Normal, Low Variance**: Participants were shown a normally distributed posterior with a standard deviation (SD) of 80 for each apartment.

**Normal, High Variance**: Participants were shown a normally distributed posterior with a SD of 250 for each apartment.

**Skew**: Participants were shown a right-skewed posterior – generated from shifted and scaled beta distribution – with a SD of 250 for each apartment.

**Bimodal**: Participants were shown bimodal posterior – generated from a mixture of two normal distributions with equal weights – with a SD of 250 for each apartment.

Note that each participant was shown the same type of (or absence of a) the posterior distribution throughout the study, as showing a participant different types of posterior distributions may affect the participant's behavior in subsequent apartments pricing questions in the study.

Participants were randomly assigned with equal probability to one of the 5 conditions. The same set of apartments was shown in each condition, although the order of the apartments was randomized. The mean of the distribution for each apartment was held fixed across conditions, and the distribution that the participant had been randomly assigned to was scaled to have that mean. The mean was marked on the visualization as the model's prediction, and it corresponds to the point prediction given in the no uncertainty case.

### 3.2.3 Procedure

At the beginning of the study, all participants were trained on how to interpret posterior distributions with a tutorial page (See Figure 1). This shows examples of different distributions of uncertainty (corresponding to the 4 conditions where participants are shown uncertainty in our experiment), and describes how to read the figures showing the uncertainty distribution. Participants are shown the same tutorial regardless of which condition they are randomized to.

To allow participants to calibrate their estimates of apartment rental prices and become familiar with the performance of the model, all participants were then taken through a practice run. Each participant was shown 3 example apartments and asked to predict their prices. Participants were told that the model slightly overestimated the true rental price in one case (prediction = \$2,644, truth = \$2,560), slightly underestimated the true rental price in one case (prediction = \$3,561, truth = \$3,601), and moderately overestimated the true rental price in one case (prediction = \$4,800, truth = \$4,290) the so that (i) participants are not incentivized to blindly agree with the ML model, and (ii) participants could not easily decide that the model always over or underestimated the true value.

After finishing this practice run, participants then completed the following task for 7 new apartments. For each apartment, the participant was asked for two estimates of rental price. Specifically, the participant was first presented with the square footage, number of rooms, and relative size of each apartment, and then asked to make a prediction about the monthly rental cost of the apartment. Then, the participant was presented with the prediction of the ML model, and the corresponding posterior predictive distribution wherever applicable. They were then asked to update their original prediction based on this information. They were reminded of their original prediction when completing this part. This design allowed us to quantify the impact of showing participants the model output by the difference between the participant's first and second estimates; This was suggested as good practice by prior work (Poursabzi-Sangdeh et al., 2021). Participants were not given feedback on whether or not they were correct after any of these trials. Also, participants were not told what features the model used to predict the price of the apartments.

At the end of the survey, participants were asked to rate their trust in and the usefulness of the model using a 5-point Likert scale (Likert, 1932). They were also asked to rate their familiarity with ML by choosing one of the following options – "No understanding or little understanding", "Worked with it a few times", "Worked with it many times or on a larger project". Participants were also asked if they ever lived in off-campus housing in Cambridge, MA or surrounding areas. The response to this question conveys participants' domain expertise in Cambridge rental listings, since people who have lived off campus in the Cambridge area are more likely to be familiar with apartment rental prices in the area than those who have not.

Participants were paid \$3 for completing the study and could earn up to an additional \$30 based on their performance in the study. Participants were told that their performance is measured based on the average distance between each apartment's true price and their first and second estimates. For the purposes of distributing bonus payments, we set the true price of the apartments to the model predictions, although recall that users were not told the true price of the 7 apartments at any point in the study. In each of the two phases of our study (see Section 3.2.5), the three participants with the lowest average distance received the \$30 bonus. The median time to complete the study was approximately 10 minutes.

### 3.2.4 Interface

The interface (see Figure 2 for an example depicting the **Normal, High Variance** condition) contains three important elements to aid participants with the study. First, it contains the model's estimate of the apartment price in bold (A). Second, if a participant was randomly assigned to an uncertainty group, the

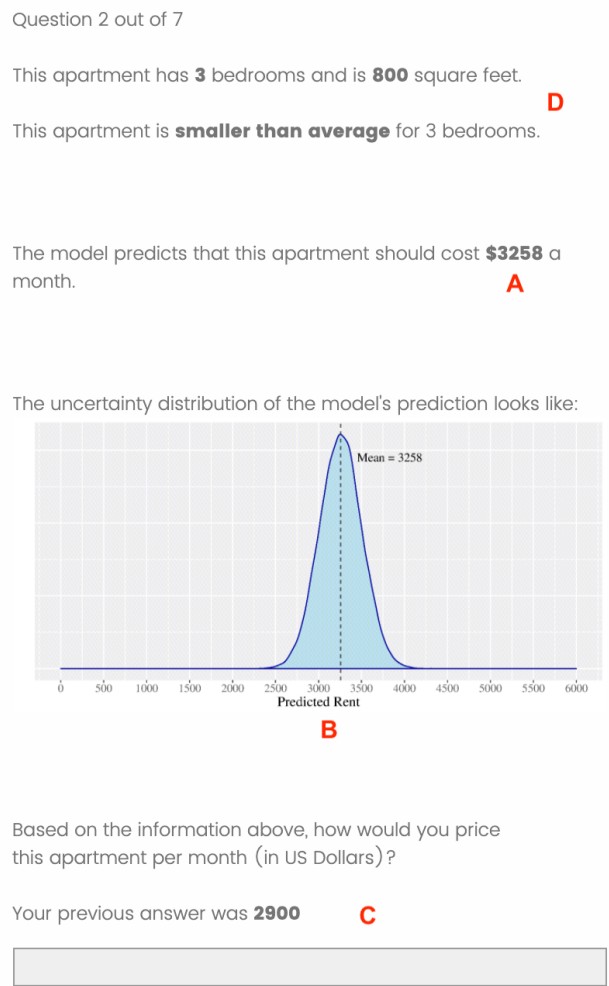

Figure 2: Example of the interface for a participant in the Normal, High Variance group after providing an initial estimate of the monthly rent of the apartment. A shows the model's prediction, B shows the visualization of the model's prediction uncertainty, C shows the participant's first response so they can reference it, and D shows the apartment's features.

interface shows the corresponding uncertainty distribution in color (B). Third, participants are reminded of their previous estimate in bold so they may choose to condition on it (C). It also contains the apartment's features (D).

### 3.2.5 Participants

In order to target participants with different degrees of ML and domain expertise for our study, we recruited participants in two ways[1]. First, we recruited 95 participants which were mainly students and researchers spanning various fields (machine learning, biology, physical sciences, history, etc.) across multiple universities. We recruited these participants by advertising in multiple mailing lists and reaching out to various faculty across multiple universities. Additionally, we recruited an additional 95 participants on the platform Prolific. We required Prolific participants to have a minimal approval rate of 98%, have completed a minimum of 250 previous submissions on the Prolific platform, and be located in the United States.

---

[1]This study was approved by our institution's IRB.

### 3.3   Analysis

We look at 3 metrics to understand the extent to which seeing predictive uncertainty affects people's decisions: the distance (in the $L_1$ norm) between the first estimate (before seeing any model outputs) and the second estimate (after seeing the prediction and uncertainty wherever applicable) – we call this the "update"; the distance between the second estimate (after seeing the model's prediction) and the model's prediction – we call this the "final disagreement"; and the distance between the first estimate (before seeing the model's prediction) and the model's prediction – we call this the "initial disagreement". The update allows us to determine the extent to which participants updated their estimate based on the model's prediction and uncertainty information. The final disagreement measures the extent to which people agree with what the model predicted. The initial disagreement allows us to understand any baseline differences between participants of varying levels of expertise before they were shown the model's prediction and uncertainty information.

Our primary analyses use mixed-effects regression models for these three metrics. Unlike regression models with only fixed effects, mixed effects models allow one to properly account for correlation between repeated measurements from participants in the study (e.g., some participants may consistently change their estimates a large amount while others may consistently change their estimates by a small amount) (Bates et al., 2015). A random intercept was included for the participant ID in all models. No other random effects were included in the models. All the relevant factors including the type of predictive uncertainty, background in ML, experience living in Cambridge, self-reported usefulness and self-reported trust in the model were included as fixed effects in their respective analyses. All p-values reported are derived from a one-way ANOVA based on these models. We report p-values that are less than 0.05 as *significant* and p-values in the interval $[0.05, 0.10]$ as *marginal*. The bar plots illustrate the sample mean of the outcome in the relevant group of participants $\pm 1$ standard error based on the fitted model.

## 4   Results

In this section, we describe in detail the results we observed. Specifically, we analyze the data along 2 dimensions: (1) how decision making differed between experts (either in the domain or ML) and non-experts, and (2) how decision making differed when shown different types of model uncertainty. We describe each of these in turn below, then run an analysis of the effect of model uncertainty on decision making stratified by participant expertise. In Appendix A, we analyze the association between subjective metrics – confidence and trust – and how users updated their estimates to agree with the machine prediction.

### 4.1   Comparing Decision Making Between Experts and Non-Experts

Our first set of research questions are about how people's willingness to update their estimates differed based on their familiarity with either the domain (apartment rentals in Cambridge, MA) or ML models. Table 1 captures the number of responses (i.e., the total number of apartment pricing examples completed) and participants from each category of ML expertise and domain expertise. Figure 3 illustrates the magnitude of the update (left), final disagreement (middle), and initial disagreement (right) among participants in each of the two categories of expertise in Cambridge apartment prices (top) and two categories of ML expertise (bottom). Throughout, we combined the ML expertise categories of "No understanding or little understanding" and "Worked with it a few times" into a single category which we refer to as "No ML background / Some ML background", and we relabelled the category "Worked with it many times or on a larger project" to "Strong ML background" for clarity.

*We found that participants with domain expertise in Cambridge apartment prices had lower initial disagreement and smaller updates compared to those without such expertise. However, the final disagreements of both groups were similar.* Participants who have previously lived in or around Cambridge made initial estimates that were closer to the model's prediction (smaller initial disagreement) even before having seen the prediction compared to those who never lived in Cambridge (significant: $F(1, 188) = 14.4761$, $p = 0.0002$). This likely reflects their increased familiarity of rental prices in Cambridge. However, we also observed that domain experts had smaller updates (significant: $F(1, 188) = 23.0224$, $p < 0.0001$), and there was no

Table 1: Number of responses and participants with each level of expertise for the domain and ML. We have a reasonable number of responses and participants with each level of expertise.

|  | Responses ($N = 1,330$) | Participants ($N = 190$) | % |
|---|---|---|---|
| **Experience living in Cambridge** |  |  |  |
| Yes | 483 | 69 | 36.32% |
| No | 847 | 121 | 63.68% |
| **Experience with ML** |  |  |  |
| Strong background | 364 | 52 | 27.37% |
| Some or no background | 966 | 138 | 72.63% |

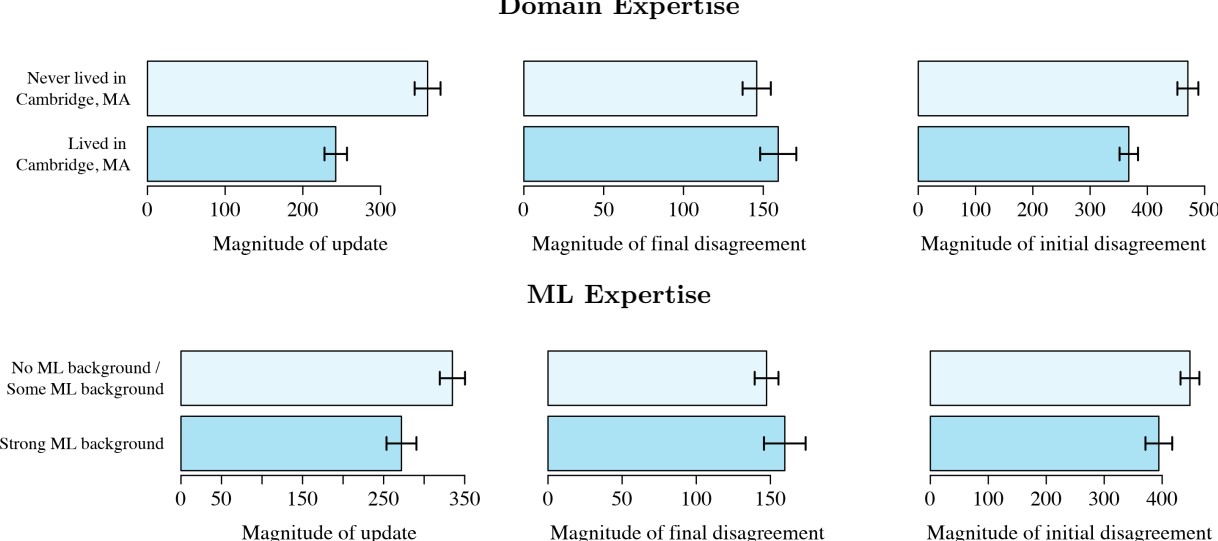

Figure 3: Magnitude of update (left), final disagreement (center) and initial disagreement (right) stratified by expertise in domain (top) and ML (bottom). Bar width corresponds to the mean, and standard errors are shown in black. Those without domain expertise had higher initial disagreement, and updated their predictions more, but they had similar final disagreement to those with domain expertise. Similar trends held for the analysis comparing ML experts and non-experts.

significant difference between domain experts and non-experts in their final disagreement with the model ($F(1, 188) = 0.8631$, $p = 0.3541$). This suggests that the ML model equalized differences in the initial disagreement between experts and non-experts by bringing participants' final disagreement with the model into the same range, regardless of expertise.

*We found that participants with more ML expertise updated their estimates less after seeing the model's prediction (significant: $F(1, 188) = 5.1232$, $p = 0.0248$).* Similar to the analysis comparing domain experts in Cambridge apartment prices to non-experts, (i) the final disagreement did not significantly differ between the ML experts and non-experts, and (ii) the ML non-experts had smaller initial disagreement (marginally significant: $F(1, 188) = 3.1833$, $p = 0.0760$). These trends may be attributed to the ML expert group having a greater portion of participants with domain expertise in Cambridge compared to the ML non-expert group. When accounting for domain expertise (i.e., by including a fixed effect term for domain expertise in our mixed-effect regression models), there were no significant differences between ML experts and non-experts in their initial disagreement, magnitude of update, or final disagreement.

### 4.2  Impact of Type of Uncertainty on Decision Making

Our second set of research questions are about the extent to which showing uncertainty influences the decisions people make, and how the type of uncertainty modulates that effect. To this end, we plot the magnitude of the update (left), final disagreement (middle), and initial disagreement (right) for each of the five conditions corresponding to a different kind of uncertainty or no uncertainty (Figure 4).

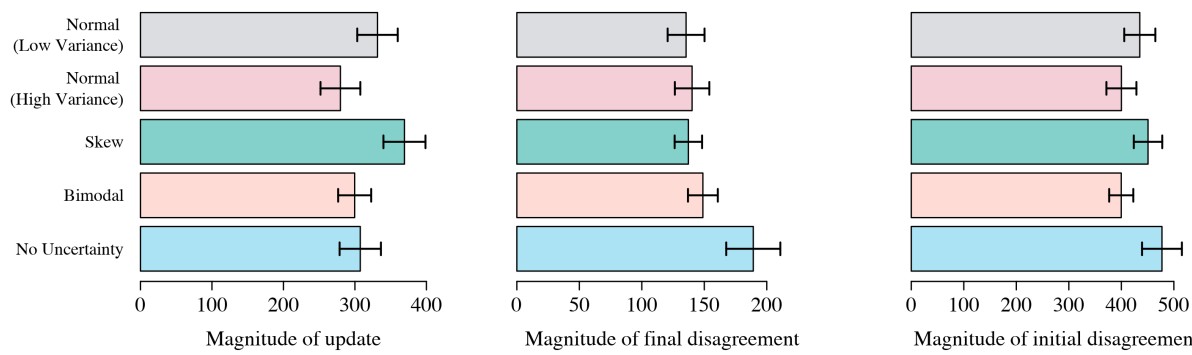

Figure 4: Magnitude of update (left), final disagreement (center) and initial disagreement (right) stratified by uncertainty condition. Bar width corresponds to the mean, and standard errors are shown in black. The final disagreement differs between conditions, with highest agreement for normal-low variance, and lowest agreement for no uncertainty.

*We found that the type of uncertainty affected participants' final disagreement with the model. Furthermore, showing no uncertainty resulted in final estimates that were farthest from model predictions.* The type of uncertainty had an effect on how closely participants' second estimate agreed with the machine's prediction (size of the final disagreement) (marginally significant: $F(4, 185) = 2.2149$, $p = 0.0690$). It can be seen from Figure 4 that the condition where participants were shown a normal distribution with low variance moved people closest to the model's prediction, however even in cases like the normal distribution with high variance where the model is clearly uncertain, or the bimodal distribution where there is very little probability mass on the model's prediction, people were still moving their second estimates closer to the model's prediction compared to when they were shown no uncertainty. More specifically, participants shown uncertainty had an average final disagreement \$49 smaller than those not shown uncertainty (significant: $F(1, 188) = 8.3989$, $p = 0.0042$).

We do not see statistically significant differences in the magnitude of the update ($F(4, 185) = 1.5239$, $p = 0.1969$). This could be tied to high variance in the original estimate before being shown the prediction, although these are also not significantly different ($F(4, 185) = 1.2734$, $p = 0.2820$), suggesting that our randomization worked.

### 4.3  Impact of Type of Uncertainty on Decision Making Stratified by Expertise

Finally, we present an analysis of the impact of uncertainty type stratified by participant expertise. The impact of the type of uncertainty on agreement with the prediction may be modulated by expertise. Table 2 shows the number of responses and participants in each uncertainty group, stratified by different levels of domain and ML expertise. While the sample sizes are small to draw strong conclusions, this analysis helps us gain further context on how expertise and uncertainty type interact, and suggests areas for future research. Figure 5 illustrates the magnitude of the update (left), final disagreement (middle), and initial disagreement (right) for each uncertainty type/no uncertainty among participants in each of the two categories of domain expertise. Figure 6 shows analogous results with respect to ML expertise.

*Uncertainty type by domain expertise: While we found that the magnitude of the update and initial disagreement were much higher for people without domain expertise, the relative trends of the impact of the*

Table 2: Number of responses and participants (within brackets) in each uncertainty group, stratified by expertise for the domain and for ML. Most groups have more than 70 responses and 10 participants.

| Expertise | No Uncertainty | Normal (Low Variance) | Normal (High Variance) | Skew | Bimodal |
|---|---|---|---|---|---|
| **Domain** | | | | | |
| Yes | 112 (16) | 105 (15) | 98 (14) | 77 (11) | 91 (13) |
| No | 168 (24) | 154 (22) | 154 (22) | 189 (27) | 182 (26) |
| **ML** | | | | | |
| Strong | 105 (15) | 63 (9) | 63 (9) | 77 (11) | 56 (8) |
| Some or no | 175 (25) | 196 (28) | 189 (27) | 189 (27) | 217 (31) |

**Domain: Expert**

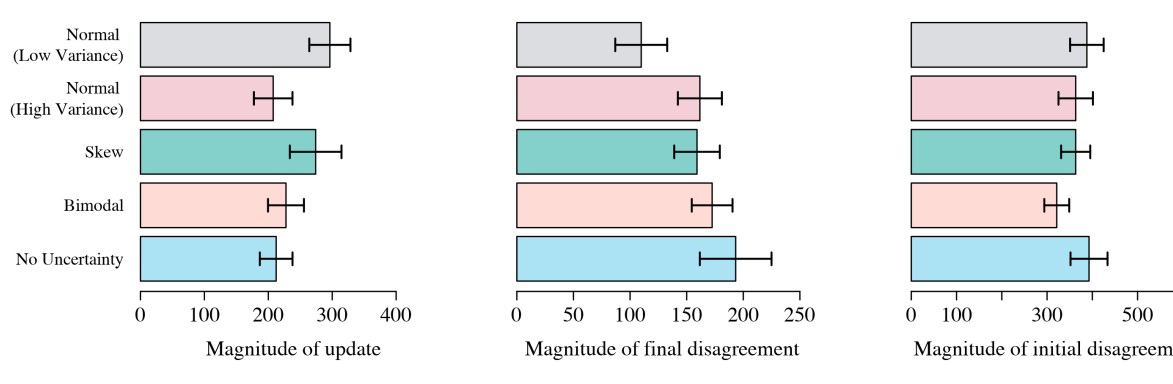

**Domain: Non-Expert**

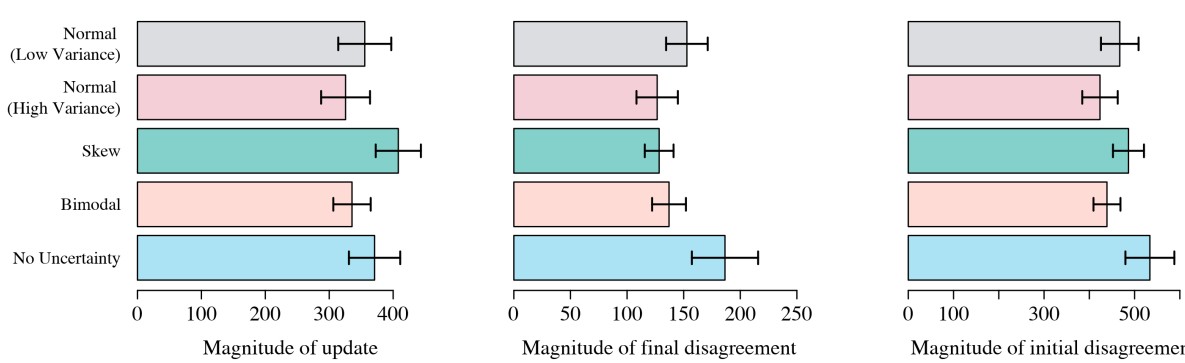

Figure 5: Magnitude of update (left), final disagreement (center) and initial disagreement (right) stratified by uncertainty condition and by domain experts (top) and non-experts (bottom). Bar width corresponds to the mean, and standard errors are shown in black. The trends in the magnitude of update across uncertainty types were similar for domain experts and non-experts, however non-experts had larger updates across conditions. Non-experts and experts had similar final disagreements.

*type of uncertainty were similar for domain experts and non-experts.* The normal low-variance and skew conditions had amongst the largest updates for both domain experts and non-experts, and the normal high variance condition had the smallest updates for both groups. For both domain experts and non-experts, the initial disagreement was similar between the different uncertainty conditions, as one would expect due to the randomization.

*Uncertainty type by domain expertise: We found that participants in the no uncertainty condition had the largest final disagreement for both domain expertise groups.* The final difference between the participants'

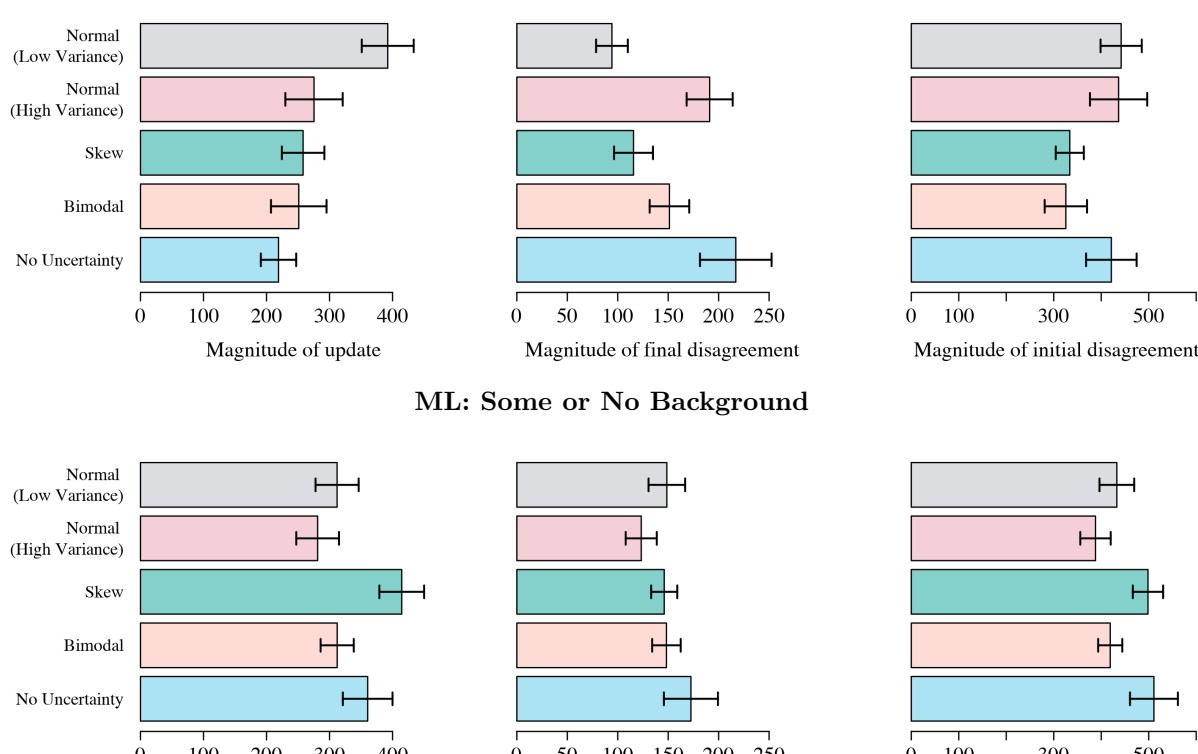

Figure 6: Magnitude of update (left), final disagreement (center) and initial disagreement (right) stratified by uncertainty condition and by background in ML. Bar width corresponds to the mean, and standard errors are shown in black. In the ML expert group, the type of uncertainty affected the magnitude of the update (significant: $F(4, 47) = 3.0754$, $p = 0.0249$) and the final disagreement (significant: $F(4, 47) = 3.6643$, $p = 0.0112$). ML experts updated their estimates the most when shown the normal low-variance uncertainty, while all other types of uncertainty had a somewhat similar effect to not showing any at all. There are no comparable marked differences for the non-experts.

estimates and the model's predictions were relatively consistent between the different types of uncertainty for both groups. For domain non-experts, the final disagreement was largest in the no uncertainty condition. While this was also true for domain experts, it was not as marked.

*Uncertainty type by ML expertise: We found that ML experts updated their original estimates more with the normal low variance condition than for any other, while there were no clear trends for ML non-experts.* From the left column of Figure 6, the normal low variance condition had much larger updates for experts (top) than those with only some ML experience or no ML experience (bottom). For the experts, the updates were much larger for this condition than for any of the others, while the updates were somewhat similar for all of the other uncertainty conditions and the no uncertainty condition. Perhaps ML experts are used to working with normal distributions and trust the prediction when the ML model is relatively certain, but do not agree to the same extent ML models that do not present their uncertainty, or present uncertainty estimates that have high variance and possibly hard-to-interpret distributions. For the non-experts, the trends were not as clear, and the updates appear relatively similar for the different uncertainty conditions. Perhaps participants with less experience with ML do not consider the variance of the distribution as much and have less of a prior on which uncertainty displays from the model are reliable. Further exploring this hypothesis is interesting future work, as it suggests that ML non-experts are perhaps less likely to respond appropriately to uncertainty.

# 5 Discussion and Conclusions

We studied how conveying predictive uncertainty to end users impacts decisions in the context of ML assisted decision making. In particular, we explored the extent to which different properties of the posterior predictive distribution affected participant agreement with the ML model's prediction in an apartment pricing task. We also explored how participant agreement with the ML model's prediction differed between experts and non-experts. Better understanding when humans agree with machine predictions, and how human expertise and machine uncertainty influence this agreement, lays the foundation for successful human-AI teamwork.

We presented new insights about how uncertainty can affect ML assisted decisions. Most interestingly, showing posterior predictive distributions (regardless of the shape and variance of the distribution) increased participant agreement with model's predictions compared to showing no uncertainty estimates at all. However, we also found that people with ML expertise agreed with the model most when provided with posterior predictive distributions that were normally distributed and had low variance. This suggests that the level of human expertise may modulate the effect of showing posterior predictive distributions, and therefore should be taken into account.

## 5.1 Limitations and Future Directions

While our study makes one of the first attempts at investigating the impact of different types of posterior predictive distributions in ML assisted decision making, it has limitations which may affect the generalizability of our findings. First, while the domain experts in our study had significantly more knowledge of apartment prices in Cambridge compared to the non-experts as evidenced by having significantly smaller initial disagreement with the model, it would be a good idea to consider users with a higher degree of domain expertise such as local real estate agents. In this case, we would expect greater differences in decision making between domain experts and non-experts. Second, as our study focuses entirely on the task of estimating apartment prices, results may differ for other decision making tasks and in other contexts. For example, human behavior studies have found that participant behavior can vary considerably between different tasks or in different contexts due to factors such as risk sensitivity (Lakkaraju & Bastani, 2020; Poursabzi-Sangdeh et al., 2021; Attali et al., 2011). Third, the two-stage nature of our study design (i.e., participants providing price estimates prior to and after seeing the model output) may be susceptible to self-priming effects (i.e., participants' price estimates after seeing the model output may be affected by having to provide initial price estimates without seeing the model output). To help mitigate potential self-priming effects, we offered a \$30 incentive for users to make the most accurate apartment price predictions. Furthermore, this study design has been suggested as good practice by prior work (Poursabzi-Sangdeh et al., 2021).

Besides the type of posterior predictive distribution and expertise of the decision makers, there may be other important factors that affect the impact of predictive uncertainty in ML assisted decision making. One such factor may be model performance. For example, if decision makers are aware that the model has excellent performance, they may heavily rely on the model regardless of their expertise or the shape or variance of the posterior predictive distribution. Exploring the impact of model performance in this context would be an interesting avenue for further work.

The setting we considered is ML assisted decision making under limited information. In this context, one can consider scenarios where users have less than, more than, or the same features as the ML model. Each of these three scenarios has been studied by seminal works in the ML assisted decision making literature. For example, Poursabzi-Sangdeh et al. (2021) studied scenarios where users have the same features as the ML model and scenarios where users have more features than the ML model. As one of the first studies investigating how different types of posterior predictive distributions affect ML assisted decision making (and how such effects vary between domain/ML experts and non-experts), we did not want to introduce additional complications arising in the scenario where users have fewer features than the ML model. For instance, if the ML model includes some additional features (not available to the user) that are highly influential for a few apartments, model predictions may diverge heavily from experts' understanding and create distrust in the model. Exploring these additional scenarios would be interesting future directions.

Finally, note that there are several sources of predictive uncertainty, which have been categorized into data uncertainty (also referred to as aleatoric uncertainty), model uncertainty (also referred to as epistemic uncertainty), and distributional uncertainty (Malinin & Gales, 2018). Prior works have highlighted the importance of these different sources of uncertainty (e.g., Dusenberry et al. (2020) on model uncertainty). While our work focused on posterior predictive distributions – which captures data and model uncertainty (Dusenberry et al., 2020) – it would be interesting to investigate the impact of presenting all these three of these types of uncertainty to decision makers.

### Acknowledgements

We would like to thank the reviewers and action editor for their comments which helped us improve the paper. SM and IL were supported by the National Science Foundation Graduate Research Fellowship Program under Grant No. DGE1745303. SM was also supported by the National Library Of Medicine of the National Institutes of Health under Award Number T32LM012411 and Fonds de recherche du Québec – Nature et technologies B1X research scholarship. HL is supported in part by NSF awards #IIS-2008461 and #IIS-2040989, and research awards from Google, JP Morgan, Amazon, Bayer, Harvard Data Science Initiative, and Dˆ3 Institute at Harvard. Any opinions, findings, and conclusions or recommendations expressed in this material are those of the authors and do not necessarily reflect the views of the funding agencies.

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

## A    Correlations Between Self-Reported Measures and Decision Making

The final question we explored was to what extent self-reported measures of trust and usefulness correlated with people's actual decisions. Table 3 captures details of participant ratings indicating how much they trust the model and how useful they find it. We combined the usefulness categories of "Very useless" and "Somewhat useless" into a single category called "Useless", and we combined the categories of "Somewhat useful" and "Very useful" into a single category called "Useful". Similarly, we combined trust categories of "None at all" and "A little" into a category called "Little/None", and we combined the categories "A lot" and "A great deal" into a category called "Lot". In the top panel of Figure 7, we give the average magnitude of update (left panel), final disagreement (middle panel), and initial disagreement (right panel) among participants in each category of self-reported trust. The bottom panel of Figure 7 gives the analogous results for self-reported usefulness.

*We found that self-reported trust correlated with the magnitude of the final disagreement (significant: $F(2, 187) = 4.5907$, $p = 0.0113$), but the trend was less clear for usefulness.* The magnitude of the final disagreement largely decreased with trust, as one may expect. While the magnitude of the update increased with trust, the trend was not significant ($F(2, 187) = 1.8490$, $p = 0.1603$). We did not observe any statistically significant differences in any of the three considered metrics between the groups defined by self-reported usefulness of the model.

Table 3: Number of responses and participants with each level of self-reported trust, and perceived usefulness of the model. Most participants trusted the model moderately or a lot, and most participants found the model useful.

| Self-reported measure | Responses ($N = 1,330$) | Participants ($N = 190$) | % |
|---|---|---|---|
| **Trust in the model** | | | |
| Little/None | 168 | 24 | 12.63% |
| Moderate amount | 616 | 88 | 46.32% |
| Lot | 546 | 78 | 41.05% |
| **Usefulness of the model** | | | |
| Useless | 70 | 10 | 5.26% |
| Neither useful nor useless | 112 | 16 | 8.42% |
| Useful | 1148 | 164 | 86.32% |

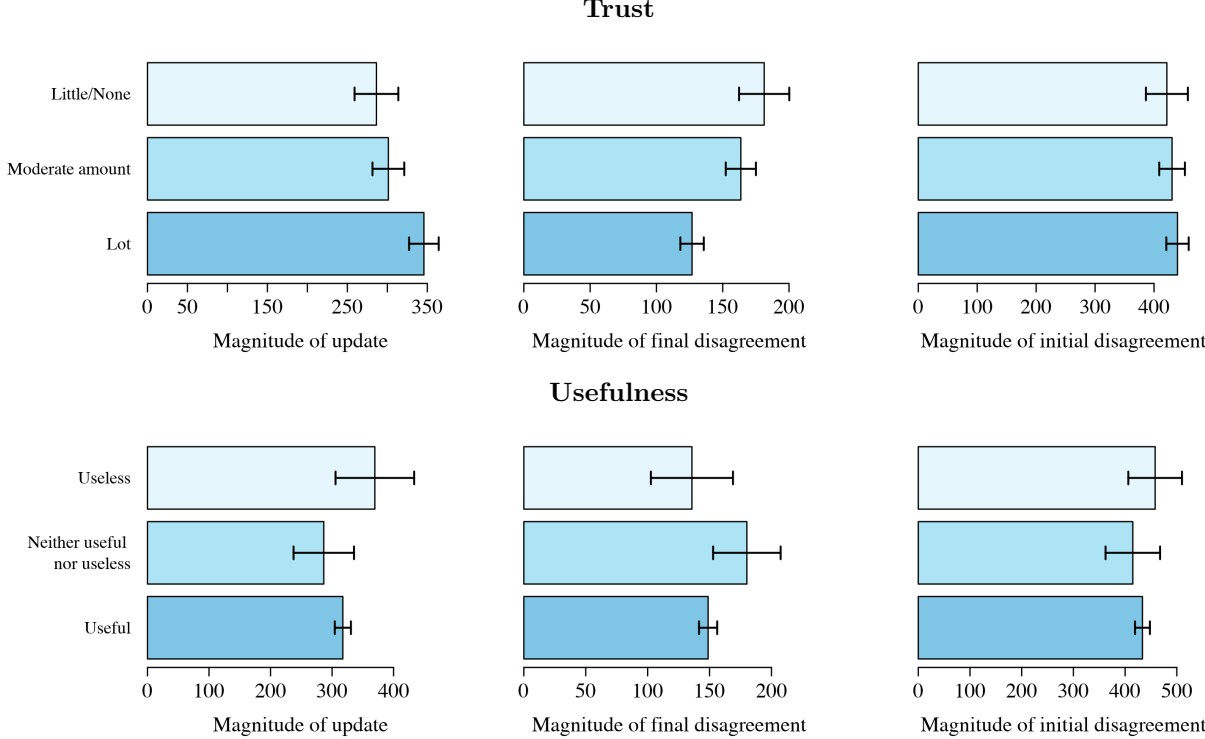

Figure 7: Magnitude of update (left), final disagreement (center) and initial disagreement (right) stratified by self-reported trust (top) and self-reported usefulness (bottom). Bar width corresponds to the mean, and standard errors are shown in black. The magnitude of the final disagreement was significantly different between the groups defined by self-reported trust. There were no significant differences between the groups defined by self-reported usefulness.

## B  Additional Study Details

### B.1  Additional Participant Details

Recall that we recruited two groups of participants in our study. The first group of participants (i.e., the students and researchers) were recruited by advertising in mailing lists and contacting faculty across multiple universities during July–September 2020. There were a total of 95 participants in this group.

The second group of participants were recruited from the platform Prolific during April 2023. A total of 95 individuals from this group completed the study. There were four participants who did not complete the study. None of their responses were included in our analyses. The average age of the study participants from Prolific was 37.14 years (range: 19 years to 66 years). A total of 41 (43.16%) of these participants were male and 54 (56.84%) were female.

### B.2  Additional Data Analysis Details

There were three cases where a participant made a clear typo in their response. In each of these cases, the participant appeared to have accidentally included an extra 0 or omitted a 0 in their apartment rental price estimate. These three cases are detailed below.

- One participant gave a second estimate of $200, which we took to be $2,000 in our analyses. This corresponded to the apartment with 1 bedroom and 500 square feet. This participant's first estimate was $1,500 and the model prediction was $2,105.

- One participant gave a second estimate of \$44,000, which we took to be \$4,400 in our analyses. This corresponded to the apartment with 4 bedrooms and 1,400 square feet. This participant's first estimate was \$4,300 and the model prediction was \$4,498.

- One participant gave a first estimate of \$200, which we took to be \$2,000 in our analyses. This corresponded to the apartment with 1 bedroom and 800 square feet. This participant's second estimate was \$2,400 and the model prediction was \$2,405.

