# OpenReview forum: "When Does Uncertainty Matter?: Understanding the Impact of Predictive Uncertainty in ML Assisted Decision Making"
_TMLR — Accepted by TMLR_

### Review · Reviewer_pcWi · 2023-03-28

**Summary Of Contributions:**

This submission studies the effects of communicating predictive uncertainty in ML-assisted human decision-making. User studies are conducted to determine the effect of different distributions of predictive uncertainty on human agreement with the ML prediction. The effects of human expertise in both ML and the application domain (apartment rent prediction) are also studied. The main findings are that showing any distribution of predictive uncertainty increases agreement with the ML prediction, with the closest agreement under low-variance normal distributions. This trend is especially noticeable for people who are not domain experts.

**Audience:**

Yes

**Claims And Evidence:**

No

**Requested Changes:**

- Weaknesses 1 (too few input features) and 3 (too few observations, which could be remedied in more than one way) are critical for me.
- Addressing weakness 2 would obviously sharpen the focus on uncertainty, but I could live without it if the authors have a good reason, given the presence of the no-uncertainty condition (especially if each participant is exposed to multiple conditions).
- Weakness 4 is concerning but not a deal-breaker.
- Addressing the minor comments and questions would strengthen the work.

**Strengths And Weaknesses:**

## Strengths
1. Systematically studies the effects of different predictive uncertainty distributions on human-ML agreement, across experts and non-experts. This distinguishes the submission from prior work.
1. The paper is very clearly written and easy to read
1. Appropriate statistical methods are used and the authors also refrain from using them and making strong claims in the "qualitative analysis" of Section 4.3 where the sample sizes are too small

## Weaknesses
1. The prediction task is based on only two input features, number of bedrooms and square footage. To me, this is too simple to represent meaningful applications of ML-assisted decision-making, where the data is high-dimensional and complex enough to motivate the use of ML models (beyond linear regression) for assistance. In short, I think two features is too few to draw generalizable insights. It also seems too simple for the specific application of rent prediction, where even unassisted humans consider many more features than number of bedrooms and square footage.
1. The experimental design somewhat confounds the effects of showing predictions and uncertainty because the two are always presented together, except for the no-uncertainty experimental condition. I thought from paragraph 1 in Section 3.1 that "agreement with the model's prediction [is measured] before and after being shown the uncertainty information" alone, which would indeed isolate the effect of uncertainty.
1. The number of participants is too small to reliably measure effects across all the dimensions considered, specifically the interaction of predictive uncertainty distribution with human expertise in Section 4.3. This is especially true of the group with no/weak ML background. It is presumably easier to recruit more of such non-ML experts.
    1. On a related note, what about exposing each participant to more than one experimental condition, and increasing the number of apartments shown accordingly? 10 apartments did strike me as low. More importantly, this could increase the overall number of observations and it might also allow for greater statistical power in comparing the conditions. The reason for the latter is that for observations from the same participant but under different conditions, the between-participant variation is removed.
1. If I understand correctly, the additional $30 payment incentivizes agreement with the ML model since the "true price" is taken to be the ML prediction. I am a bit concerned that this incentive biases all the measurements of agreement.

## Minor Comments and Questions
- Introduction, paragraph 1: "complimentary" --> "complementary"
- Introduction, car selling example: This example may be complicated by the game theory aspect, i.e., the best asking price may not equal the predicted selling price depending on how prospective buyers may respond.
- Section 3.1, paragraph 4: The wording "when that uncertainty is larger and encompasses more plausible predictions vs. when it is smaller and more targeted" suggests to me that there is also a choice of range over which to show uncertainty (e.g. choosing lower and upper quantiles that bound a range, or the width of a prediction interval), in addition to the intrinsic variance of the distribution. But the former choice is not considered in this paper so I would suggest rephrasing.
- For the bimodal distribution, how far apart are the two means and what are the two individual standard deviations? Giving the overall SD does not fully specify it.
- How long did the participants take on average to complete the study?
- Figure 2: The sentence "this apartment is smaller than average for 3 bedrooms" is technically additional information beyond the two feature values (specifically information about the conditional distribution of square footage).
- Section 3.3: Is the random intercept for participant ID the only random effect in the mixed-effects model?
- Figure 3 caption: "without domain expertise have lower initial disagreement" --> "higher"

---

> ### Author Response · Authors · 2023-04-22
> **Response to Reviewer pcWi (Part 1 of 2)**
>
> Dear Reviewer pcWi,
>
> Thank you for your helpful review. Below we respond to each point in turn.
>
> The revised paper marks the changes made in blue text.
>
> ## Weaknesses
>
> **Comment 1**:
> >The prediction task is based on only two input features, number of bedrooms and square footage. To me, this is too simple to represent meaningful applications of ML-assisted decision-making.
>
> The setup of our user study does not mandate that the ML model is only based on two features. That is, we did not tell the participants which features the ML model uses to estimate apartment prices. Our study only mandates that participants are given these two features when making their estimates.
>
> In real applications of ML assisted decision making, people often have to make decisions with limited information. To provide users with relevant features, we reduced the feature space by fixing the location of the apartments to be within City A and then chose the two most important features affecting apartment prices.
>
> We revised the Introduction (Section 1, paragraph 4) and Section 3.2.1 (paragraph 1) to clarify the context of our study, i.e. of ML assisted decision making under limited information. We also revised Section 3.2.3 (paragraph 3) to clarify that the posterior distributions shown to users were not necessarily from an ML model based on these two features alone.
>
> **Comment 2**:
> >The experimental design somewhat confounds the effects of showing predictions and uncertainty because the two are always presented together, except for the no-uncertainty experimental condition.
>
> We would like to reiterate that our study design which compares participants randomized to the uncertainty groups to the no uncertainty group provides an unbiased (unconfounded) estimate of the effect of being shown uncertainty measures. An approach that additionally asks users to provide an estimate after seeing the point estimate (before the uncertainty) would also capture the impact showing uncertainty. However, a potential disadvantage of this approach is that it is more susceptible to self-priming effects, where participants may be less willing to change their estimates due to providing previous estimates.
>
> **Comment 3**:
> >The number of participants is too small to reliably measure effects across all the dimensions considered.
>
> Thank you for raising this concern. We made the following revisions to address this comment:
>
> * We re-launched our user study (on the platform Prolific) and consequently doubled the number of participants. Our study now contains 1,330 responses from 190 participants.
> * We reduced the number of ML expertise groups from three (no/weak background, some background, strong background) to two groups (no/some background vs strong background) to increase our effective sample size in the stratified analyses, as the trends in the stratified analyses were similar between the "no/weak background" group and the "some background" group.
>
> >What about exposing each participant to more than one experimental condition, and increasing the number of apartments shown accordingly?
>
> We were concerned that exposing participants to different experimental conditions may affect their behavior in subsequent questions in the study. For instance, suppose that a participant is shown the normal low variance distribution for the first apartment and is only shown the point estimate for the second apartment. Our concern is that, for the second apartment, the participant may be influenced by the posterior distribution shown for the first apartment, such as believing that the model is generally highly certain of its estimates. We clarified this point in Section 3.2.2 of the revised paper.
>
> We limited the number of apartments in the user study to 10 in order to ensure that participants are engaged throughout the entire survey. In a pilot version of the study, participants reported losing focus after approximately 10 apartments due to the repetitive nature of the study.
>
> **Comment 4**:
> > If I understand correctly, the additional $30 payment incentivizes agreement with the ML model since the "true price" is taken to be the ML prediction. I am a bit concerned that this incentive biases all the measurements of agreement.
>
> Participants were not told that the true price (in the testing examples) is the ML prediction.
>
> In fact, in the training examples where participants were shown their error from the true price, the true price was not set to the ML prediction in order to not incentivize agreement with the ML model. Specifically, in the first training example, the ML prediction was 2,644 USD and the true price was 2,560 USD; In the second example, the ML prediction was 3,561 USD and the true price was 3,601 USD; In the third example, the ML prediction was 4,800 USD and the true price was 4,290 USD. In the testing examples, users were not told the true price at all. We revised paragraphs 2 and 5 in Section 3.2.3 to clarify this point.

---

> > ### Author Response · Authors · 2023-04-22
> > **Response to Reviewer pcWi (Part 2 of 2)**
> >
> > ## Minor Comments and Questions
> > >"complimentary" --> "complementary"
> >
> > This is now fixed.
> >
> > >The wording "when that uncertainty is larger and encompasses more plausible predictions vs. when it is smaller and more targeted" suggests to me that there is also a choice of range over which to show uncertainty in addition to the intrinsic variance of the distribution. But the former choice is not considered in this paper so I would suggest rephrasing.
> >
> > We revised this sentence. It now reads as: "We wish to answer the question of how much uncertainty affects people's decisions when the variance of the posterior distribution is larger vs. when it is smaller"
> >
> > > For the bimodal distribution, how far apart are the two means and what are the two individual standard deviations?
> >
> > Letting $\mu$ denote the overall mean of the bimodal distribution, the means of the two normal distributions were approximately $\mu \pm 246$ and their standard deviations were each approximately 42
> >
> > >How long did the participants take on average to complete the study?
> >
> > The median time to completion was approximately 10 minutes. This is now clarified in the text (Section 3.2.3, final paragraph).
> >
> > >Figure 2: The sentence "this apartment is smaller than average for 3 bedrooms" is technically additional information beyond the two feature values.
> >
> > This point is now clarified throughout the paper. For example, the second sentence in paragraph 1 of Section 3.2.1 previously read as: ``Participants were provided with the number of square feet and number of bedrooms of each apartment". It now reads as: "Participants were provided with the number of square feet, number of bedrooms, and the relative size of each apartment (i.e., whether the apartment size is smaller than average, average, or larger than average for its number of bedrooms)".
> >
> > >Is the random intercept for participant ID the only random effect in the mixed-effects model?}
> >
> > Yes. This is now clarified in the text (Section 3.3, final paragraph).
> >
> > >"without domain expertise have lower initial disagreement" --> "higher"
> >
> > This is now fixed.
> >
> > Best regards,
> > The authors

---

> > ### Comment · Reviewer_pcWi · 2023-05-05
> > **my concern regarding too few features still stands**
> >
> > Thanks very much to the authors for your responses, and especially for doubling the size of the user study in a short amount of time.
> >
> > I am afraid however that my weakness 1 (too few features) still stands and is a critical issue for me. While I accept the arguments that participants were not told which features the ML model uses and that humans often have limited information (e.g. because of cognitive limitations), the fact remains that in this experiment, the ML model also uses only two features. I cannot shake the opinion that this is far too few to represent a realistic application calling for ML modeling. I do not see why the ML model, even in this rental price prediction experiment, could not be based on more features. That would actually test the stated scenario in which humans can only use limited information while ML models can take more into account. Doing so may introduce aspects that would increase the realism in my view, such as making the ML model significantly more accurate than humans (at least where data coverage is good), and/or having ML predictions that vary in unexpected ways from the humans' perspective.
> >
> > Follow-ups on my other major comments:
> > - Comment 2: I think it may still be of interest to try asking for user estimates after seeing the point estimate and before the uncertainty. This might show the extent of the self-priming effect, and if self-priming is not significant, then it could provide a lower-variance estimate of the effect of showing uncertainty because the comparison is within subjects rather than across subjects.
> > - Comment 3: Thank you for also explaining the limitation of each participant to one experimental condition and to 10 apartments.
> > - Comment 4: Since participants were told that they would be judged based on the true price, I do not understand why bonus payments were not awarded on this basis rather than agreement with the ML model.

---

> > > ### Author Response · Authors · 2023-05-07
> > > **Additional Response to Reviewer pcWi**
> > >
> > > We thank the reviewer for their follow-up comments. Below we address these comments:
> > >
> > > > ML model also uses only two features. I cannot shake the opinion that this is far too few to represent a realistic application calling for ML modeling
> > >
> > > The setting we consider is ML assisted decision making under limited information. In this context, one can consider scenarios where users have less than, more than, or the same features as the ML model. Each of these three scenarios has been studied by seminal works in the ML assisted decision making literature. For example, Poursabzi-Sangdeh et al. (2021) studied scenarios where users have the same features as the ML model and scenarios where users have more features than the ML model (in which case their ML model is also based on just two features). Kleinberg et al. (2017) consider scenarios where users have more features than the ML model.
> > >
> > > As one of the first studies investigating how different types of posterior predictive distributions affect ML assisted decision making (and how such effects vary between domain/ML experts and non-experts), we did not want to introduce additional complications arising in the scenario where users have less features than the ML model. For instance, if the ML model includes some additional features (not available to the user) that are highly influential for a few apartments, model predictions may diverge heavily from experts’ understanding and create distrust in the model.
> > >
> > > We agree with the reviewer that exploring these additional scenarios would be interesting future directions. We will add a paragraph in the Discussion describing these additional scenarios one may consider.
> > >
> > > Additionally, we would like to clarify that the posterior distributions shown to users in this study were not generated by an ML model and were instead hand crafted since the goal was to understand how users would respond to different shapes of posterior distributions (See paragraph 3 of Section 3.2.1 for more details). The different types of posterior distributions we considered are widely encountered in Bayesian ML models (Gelman et al. 2013).  A strength of our study design is that the posterior distributions are invariant to the number of features used by the ML model. We hope that this helps alleviate some of your concerns about the number of features used in the ML model.
> > >
> > > Finally, we would like emphasize that, while exploring these additional scenarios would be interesting, such considerations are not related to the correctness of our study.
> > >
> > > >Thank you for also explaining the limitation of each participant to one experimental condition and to 10 apartments.
> > >
> > > We would like to clarify that assigning each participant to one experimental condition is not a limitation, but is in fact a strategy to avoid contamination where each user sees multiple posterior distribution shapes (conditions) and there might be potential interference between these conditions which in turn affect user decisions. Prior works have employed a similar strategy to mitigate these issues.
> > >
> > > > Since participants were told that they would be judged based on the true price, I do not understand why bonus payments were not awarded on this basis rather than agreement with the ML model.
> > >
> > > The apartments shown to users were hypothetical apartments, as discussed in Section 3.2.1. Consequently, we had to award participants based on the closeness of their estimates to the ML prediction. As the reviewer mentions, the participants were told that they would be judged based on the true price in order to not incentivize blind agreement with the model
> > >
> > > **References**:
> > >
> > > Forough Poursabzi-Sangdeh, Daniel G Goldstein, Jake M Hofman, Jennifer Wortman Wortman Vaughan, and Hanna Wallach. 2021. Manipulating and Measuring Model Interpretability. In Proceedings of the 2021 CHI Conference on Human Factors in Computing Systems (CHI '21). Association for Computing Machinery, New York, NY, USA, Article 237, 1–52. https://doi.org/10.1145/3411764.3445315
> > >
> > > Jon Kleinberg, Himabindu Lakkaraju, Jure Leskovec, Jens Ludwig, Sendhil Mullainathan, Human Decisions and Machine Predictions, The Quarterly Journal of Economics, Volume 133, Issue 1, February 2018, Pages 237–293, https://doi.org/10.1093/qje/qjx032
> > >
> > > Gelman, Andrew, John B. Carlin, Hal S. Stern, David B. Dunson, Aki Vehtari, and Donald B. Rubin. Bayesian data analysis. CRC press, 2013.

---

### Review · Reviewer_s9ZU · 2023-04-04

**Summary Of Contributions:**

The paper investigates to which degree humans, when predicting rental prices of apartments, adjust their predictions when being presented with the prediction of a ML model. Importantly, the paper studies the effect of showing the ML model prediction with different degrees of certainty, represented by different distributions around the mean prediction. The main finding is that showing the model’s uncertainty estimates, instead of just the mean, has an effect on the degree to which humans adjust their predictions. The study also finds that the adjustment of predictions is modulated by the shape of the distribution (e.g., low-variance Normal leads to lager adjustment compared to high-variance Normal). The results are based on a study with 95 participants, that each predicted the rental price of 7 apartments (knowing only the size and number of bedrooms). Additionally, the paper investigates whether having more or less domain knowledge has an influence on the prediction adjustments; and similarly for having little, moderate, and significant experience with ML.

**Audience:**

Yes

**Broader Impact Concerns:**

No concerns.

**Claims And Evidence:**

No

**Requested Changes:**

**Improvements / Major comments:**

1. Instead of reporting all measurements (initial and final disagreement and magnitude of update) in absolute values (dollars), report results in terms of quantities normalized by the true prize of the apartment. E.g., if the initial prediction is 1200USD for an apartment that costs 1000USD, report the initial disagreement as 1.2. This ensures that predictions and adjustments for apartments of very different sizes (and thus absolute rent) are comparable. A side-effect might be that some of the stat. tests are no longer stat. significant - which I would take as a strong hint that the effect is too weak or the sample size is too low.

2. Record more participants. Since recording for this study is neither time-consuming nor expensive (compared to typical behavioral studies), I suggest to double or triple the number of participants. Without additional data, the paper is left with the results in Fig. 3 (which look valid, but do not show the paper’s main point of investigation: different predictor distributions), and the results in Fig. 4, but since Fig. 3 and Fig. 5 suggest that experts and non-experts update differently, the stratified results are very important.

3. I think Fig. 5 is the main figure of the current study (because Fig. 4 aggregates over potentially different update mechanisms between experts and non-experts). Unfortunately it has some shortcomings: (A) within each group there is no reason to expect a difference in initial agreement (right column), but there is clearly one for the non-experts (blue bar). This is a very strong hint that the sample size for each stratified cohort is too low. (B) The largest adjustment is seen for the non-experts in the grey condition (Normal, low variance) - but this is based on 3 participants only according to Table 2. That number is concerningly low. Both issues A and B can be easily fixed by increasing the number of participants.

4. No actual expert group. The current study would be much stronger with an actual expert group (ideally recruited from local property agents). While I do not consider this a hard requirement for publication, I think it would make the work much stronger. Alternatively the task could also be switched to something where it is easier to recruit experts and non-experts from a student/faculty population.

5. Generality of findings needs to be toned down. The current work presents limited results on a single task, but is written as if findings were to hold in general. This is unjustified. Particularly with human prediction and decision-making, there is a long history of behavioral studies that show that behavior of the same participant can change drastically across tasks or even framing the same task differently (e.g. due to priming effects, risk-sensitity, etc.). Ideally, the work would present results that hold across a range of tasks (the same participants could easily be recorded to answer more questions from different tasks). I would be willing to accept a paper with a single task only, if executed very well and if these limitations are clearly discussed.

6. Add a limitations section. Provide a critical summary of the claims that can be made given the study. Clearly state and discuss current limitations and unexplained results (or results hypothesised to result from too small sample size). Discuss how this affects the generality of findings.

7. Ablations: show that there is no difference in final disagreement due to the two-stage process (i.e. no self-priming by having to give an answer before seeing the prediction). This can easily be done by having one group which sees the model prediction right away before having to make a prediction. There should be no stat. significant difference in final disagreement between this control group and the standard group; otherwise having to make a prediction first becomes essential/detrimental to the process and would need to be factored in. I find this point important and would certainly add it if additional data recording is done, but since it is somewhat tangential to the question of different uncertainty representations, I would give this point lowest priority

**Minor comments:**

The following sentence is a bit ambiguous: “Specifically, for each participant, we computed the average distance between each apartment’s true price (i.e. the model’s prediction) and the participant’s first and second estimates.”. I guess participants were only instructed to match the “true price” and not the model’s prediction (which they saw in the trials and could have trivially matched) - the authors used the model’s prediction rather than the actual true price (a month later) to compute the bonus payouts only. Is this correct?

Minor point for discussion: It is unclear that, when asking humans for a number, humans report their mean estimate (they might have a skewed distribution and report the median) - similarly, after seeing the model’s prediction humans might put more emphasis on the median and not the mean. It is thus unclear whether humans should always be expected to adjust towards the mean (e.g. in the bimodal case).

Ideally, provide some measure of quality/accuracy of the ML predictions (if the model performs badly, it would be unsurprising that experts do not update their predictions much compared to non-experts; the maximal version here would be to conduct an additional ablation study with a good model and a bad model to show that experts can spot the bad model and adjust less and non-experts cannot).


**Strengths And Weaknesses:**

**Strengths:**
 * Timely and important problem: how should ML model outputs be presented to users, and what impact do different ways of communicating predictions to users have?
 * Well written paper
 * A number of clearly stated hypotheses that are tested

**Weaknesses:**
 * Number of participants and/or apartments rated too low. Though aggregate results are technically speaking stat. significant, the stratified results in Fig. 5 (where experts and non-experts are shown independently) suggest experts and non-experts differ in their adjustments. Unfortunately the stratified cohorts in Fig. 5 are much too small (e.g., 6 or even just 3 participants in the Non-Expert category for some distributions). See Requested Changes for a more detailed discussion, but the effects and discrepancies in Fig. 5 cannot be ignored solely with the argument that aggregate statistics in Fig. 3 yield a good enough p-value. Since recording additional participants in the study is very quick and cheap (3$ per participant for rating 7 apartments) I strongly recommend collecting at least two, to three times more data.
 * Generalizability of findings is very limited. The paper is written as a general investigation into how different types of uncertainty of ML predictors affect human predictions that get to work with the model outputs. Main claims in the paper are stated very generally. This is in stark contrast to the experimental results, which investigate a single, very simple task, and 5 kinds of ML model uncertainty types (4 hand-crafted distributions). This needs to be addressed by toning down the claims and adding a well written limitations section or, even better, adding more different tasks to the study.
 * No true expert group. The expert group in the study is only marginally more experienced in the task than the non-experts. The study would greatly benefit from including another group that is recruited from local property agents (specialising in rental properties). I would not be surprised if these actual experts, who have high confidence in their predictions, would adjust a lot less based on seeing the model’s predictions - producing a much more pronounced difference between experts and non-experts and maybe being less sensitive to the type distribution of the model prediction.

**Overall verdict:** The problem that the paper contributes to answering is timely and important, and the main idea behind the study design in the paper is good. I personally would interpret the current results as a promising proof-of-concept (where the results for the main hypotheses look promising, but some effects in the stratified data can currently not be explained and are likely due to low sample size, but this needs to be investigated since these effects bleed into the aggregate data). The next step after a proof-of-concept is to conduct a larger scale study, and I think the work is ready to do this (and compared to many other behavioral studies, collecting more data is relatively quick and cheap in this case). The paper would greatly benefit from this; the findings presented would be more reliable and convincing. To me personally, the work currently does not pass the threshold for publication - though the paper is well written and the main approach sound, there is more needed than “just getting the p-values for the aggregate results right”. Some of the claims and findings in the paper are not reliable enough, and the unexplained effects (which the paper itself attributes most likely to having too small a sample size) need to be addressed. Luckily this can be done very easily and straightforwardly by collecting more data. I think there is good potential for conducting a well-rounded, exemplary empirical study, but the current state of the empirical work falls short of the high standards needed for good behavioral studies with humans. I want to strongly encourage the authors to spend the extra work to make this a strong behavioral study that stands the test of time.

---

> ### Author Response · Authors · 2023-04-22
> **Response to Reviewer s9ZU (Part 1 of 2)**
>
> Dear Reviewer s9ZU,
>
> Thank you for your thoughtful review and constructive feedback. Below we respond to each point in turn.
>
> The revised paper marks the changes made in blue text.
>
> ## Major Comments
>
> **Comments 1-3**:
>
> > Record more participants. I suggest to double or triple the number of participants.
>
> Thank you for bringing to our attention the various issues arising from having too small sample sizes. We made the following revisions to address these issues:
> * We re-launched our user study (on the platform Prolific) and consequently doubled the number of participants. Our study now contains 1,330 responses from 190 participants.
> * We reduced the number of ML expertise groups from three (no/weak background, some background, strong background) to two groups (no/some background vs strong background) to increase our effective sample size in the stratified analyses, as the trends in the stratified analyses were similar between the "no/weak background" group and the "some background" group.
>
> By making these changes to increase our sample sizes, we have updated our findings and claims throughout the paper. We also resolved the following limitations pointed out in our original study:
>
> > Within each group in Fig 5, there is no reason to expect a difference in initial agreement, but there is clearly one for the non-experts. This is a very strong hint that the sample size for each stratified cohort is too low.
>
> The magnitude of the initial disagreement is now similar across all uncertainty groups in Figure 5, as one would expect due to randomization.
>
> > The largest adjustment in Fig 5 is seen for the non-experts in the grey condition (Normal, low variance) - but this is based on 3 participants only. That number is concerningly low.
>
> The number of domain non-experts in the normal low variance condition increased from 3 to 22, as most of the additional participants we recruited were non-experts.
>
> Next, we would like to respond to the final comment relating to small sample sizes:
>
> >Report results in terms of quantities normalized by the true prize of the apartment... A side-effect might be that some of the stat. tests are no longer stat. significant - which I would take as a strong hint that the sample size is too low.
>
> We would like to clarify that all participants in our study were shown the same set of apartments.  That is, no experimental group in our study had more/less expensive apartments than the other groups. We considered normalizing the quantities of interest by the "true" apartment prices, and we found that trends were similar and in some cases even found smaller p-values when performing normalization. However, we find the raw (unnormalized) differences to be more easily interpretable in the context of our study.
>
> **Comment 4**:
> > The current study would be much stronger with an actual expert group (ideally recruited from local property agents).
>
> We agree that it would be a good idea to consider users with a higher degree of domain expertise. However, those classified as domain experts in our study did, in fact, show significantly greater knowledge in apartment prices compared to those classified as nonexperts. For instance, the domain experts made initial estimates that were closer to the model's prediction (smaller initial disagreement) before having seen the prediction compared to those classified as nonexperts. Specifically, the initial disagreement of the domain experts was on average 103 USD smaller than that of the nonexperts ($p = 0.0002$).
>
> We added two sentences in the Discussion (Section 5.1, paragraph 1) acknowledging that it would be good to consider those with a higher degree of domain expertise.

---

> > ### Author Response · Authors · 2023-04-22
> > **Response to Reviewer s9ZU (Part 2 of 2)**
> >
> > **Comment 5**:
> > >Generality of findings needs to be toned down
> >
> > Thank you for drawing our attention to this. Throughout the paper, we revised the description of our findings to more precisely reflect the results of our study. For example, the summary of our findings in the abstract was previously:
> >
> > >In this work, we carry out user studies to systematically assess how people with differing levels of expertise respond to different types of predictive uncertainty i.e., posterior predictive distributions with different shapes and variances, in the context of ML assisted decision making. Our results demonstrate that showing uncertainty information leads to smaller disagreements with the machine, regardless of the type of uncertainty, but that these effects are sensitive to expertise in both ML and the domain.
> >
> > and now reads as (changes in bold text):
> >
> > >In this work, we carry out user studies **(1,330 responses from 190 participants)** to systematically assess how people with differing levels of expertise respond to different types of predictive uncertainty i.e., posterior predictive distributions with different shapes and variances, in the context of ML assisted decision making **for predicting apartment rental prices**. **We found that showing posterior predictive distributions** led to smaller disagreements with the **ML model's predictions**, regardless of the **shapes and variances of the posterior predictive distributions we considered**, and that these effects **may be** sensitive to expertise in both ML and the domain.
> >
> > Additionally, we added a comprehensive limitations section (Section 5.1), which discusses how several limitations of our study may affect the generalizability of our findings. Further details are in our response to the following comment.
> >
> > **Comment 6**:
> > >Add a limitations section.
> >
> > We added a limitations section in the Discussion (Section 5.1) in response to several comments we received. Specifically, we discuss the following topics:
> > * Including users with a higher degree of domain expertise may result in greater differences in decision making between domain experts and nonexperts.
> > * Results may differ for other decision making tasks and in other contexts.
> > * Potential self-priming effects may result in participants updating their estimates less due to the two-stage process.
> > * Model performance may also strongly affect ML assisted decision making (including the impact of presenting posterior predictive distributions).
> > * Presenting other notions of uncertainty may also strongly affect ML assisted decision making (including the impact of presenting posterior predictive distributions).
> >
> > **Comment 7**:
> > > Ablations: show that there is no difference in final disagreement due to the two-stage process
> >
> > Our study design which requires participants to give two estimates -- one prior to seeing the model output and one after seeing the model output -- follows prior studies which had this design (Poursabzi-Sangdeh et al., 2021). Nevertheless, we were concerned about potential self-priming effects. For this reason, we put a \$30 incentive for users to make the most accurate apartment price predictions to help avoid the scenario where participants are reluctant to update their estimates.
> >
> > We added a few sentences in the Discussion (Section 5.1, paragraph 1) to acknowledge potential self-priming effects.
> >
> > ## Minor Comments
> >
> > >The authors used the model’s prediction rather than the actual true price (a month later) to compute the bonus payouts only. Is this correct?
> >
> > Correct. In the testing examples (used to compute bonus payments), we set the ``true price" of the apartments to be the model prediction just to compute the bonus payment. Participants were not told that the true price of the apartments in the testing examples is the ML prediction.
> >
> > In the training examples (not used to compute bonus payments), participants were shown the "true price" after providing both estimates (see paragraph 2 of Section 3.2.3 for details). In these training examples, we set the true price so that the ML prediction sometimes overestimated and true price and sometimes underestimated the true price to ensure that (i) participants are not incentivized to blindly agree with the ML model, and (ii) participants could not easily decide that the model always over or underestimated the true value.
> >
> > We revised paragraphs 2 and 5 in Section 3.2.3 to clarify this point.
> >
> > > Ideally, provide some measure of quality/accuracy of the ML predictions
> >
> > In our study, all participants were shown the accuracy of the model predictions in three training examples (see paragraph 2 of Section 3.2.3 for details). We agree that results may differ if users are told that the model is highly accurate versus highly inaccurate. We added a paragraph in Discussion to acknowledge that model performance may also be an important factor in ML assisted decision making (Section 5.1, paragraph 2).
> >
> > Best regards,
> > The authors

---

> > > ### Comment · Reviewer_s9ZU · 2023-04-24
> > > **Thank you for addressing the concerns raised and, most importantly, collecting additional data**
> > >
> > > I want to thank the authors for their detailed response and taking on the effort to double the number of participants within the short time-frame of the rebuttal. To me, all issues raised by the reviewers have been addressed (or clarified, like my major comment 1) - and though some issues have been left as future work, these issues feature in the discussion/limitations and the generality of the findings has been phrased appropriately. The increased participant numbers have confirmed the main findings, and have lifted the statistics on more reliable footing - thank you again for conducting the additional study; I think it has paid off.
> > >
> > > I currently have no objections to publishing the paper, but I am keen to hear the updated verdict of the other reviewers.

---

> > > > ### Author Response · Authors · 2023-04-28
> > > > **Thank you for your positive response**
> > > >
> > > > We thank the reviewer for considering our rebuttal, and for the positive feedback. We are very glad to hear that the reviewer has no objections to publishing our work. Thank you once again for all your inputs and feedback which helped us immensely in improving our work.

---

### Review · Reviewer_m3a9 · 2023-04-08

**Summary Of Contributions:**

This work conducts a set of experiments to answer the question "How do people respond to model uncertainty?" in multiple dimensions such as the effect of expertise on the task/ML, the effect of distribution type and variance. The paper uses apartment rental price regression as the main task, and recruits 95 people to make an initial prediction, see the model output (along with its uncertainty), and update their prediction. The paper collects the survey results and conducts multiple analyses.

**Audience:**

Yes

**Claims And Evidence:**

Yes

**Requested Changes:**

- If the paper could include a limitation section to address the above concerns, that would make this paper stronger.

**Strengths And Weaknesses:**

Strengths:
- According to the paper, this is the first attempt to address the question "How does various factors (e.g. distribution type, variance, human expertise) affect how people agree with model prediction?". The paper uses a well thought-out procedure to eliminate the factor of model prediction performance, and only focuses on the question. While this seems like a strength on the one hand (i.e. simplifies the question and the complexity of the required analysis), it also sets a hard limitation to the depth of the research, which I'll elaborate in the Weaknesses.
- The analysis of the results is overall quite reasonable, and the paper does a great job of summarizing the entire experiment in a few simple statements.

Weaknesses:
- As noted above, this paper simplifies the question "how does uncertainty affect human decision?" by eliminating the factor of model performance from the entire process. In other words, this paper only focuses on how distribution types and the magnitude of variance affects human decision, coupled with human expertise. To guarantee this, during the practice run, the participants were shown three model predictions where the model under-estimates and over-estimates at least one time respectively, so that people would not blindly believe the model predictions during the actual run. Although a clever approach, this seems to over-simplify how people actually engage with ML models in the real world. I believe model performance would play a crucial role in how people react to model uncertainty. For example, with a prediction model that people know to possess poor accuracy, I assume human expertise would play a much bigger role than distribution types or variance magnitude. With a very accurate model (better than most domain experts), maybe none of the factors (human expertise, distribution types, variance magnitude) would play a significant role.
- Another aspect this paper do not address in terms of model uncertainty is different types of uncertainty: epistemic uncertainty (which is also called model uncertainty), aleatoric uncertainty (a.k.a. data uncertainty). Following this nomenclature, this paper is in fact, studying data uncertainty, not model uncertainty. Some works [1] even go as far as counting out-of-distribution (distributional uncertainty) as a third type of uncertainty. If people are given all two (or three) types of uncertainty information, they would behave very differently compared to when given only aleatoric uncertainty (as this paper). For example, in safety critical tasks such as medical prediction, epistemic uncertainty could play as big a role as aleatoric uncertainty [2].

References:
[1] Malinin, Andrey, and Mark Gales. "Predictive uncertainty estimation via prior networks." Advances in neural information processing systems 31 (2018).
[2] Dusenberry, M.W., Tran, D., Choi, E., Kemp, J., Nixon, J., Jerfel, G., Heller, K. and Dai, A.M., 2020, April. Analyzing the role of model uncertainty for electronic health records. In Proceedings of the ACM Conference on Health, Inference, and Learning (pp. 204-213).

---

> ### Author Response · Authors · 2023-04-22
> **Response to Reviewer m3a9**
>
> Dear Reviewer m3a9,
>
> Thank you for your thoughtful comments. As suggested, we added a limitations section in the Discussion to address these two concerns (see Section 5.1, paragraphs 2 and 3). We discuss how model performance may play an important role in how people react to model uncertainty. We also discuss the potential impact of other types of uncertainty (e.g., distributional uncertainty) and added the given references.
>
> The revised paper marks the changes made in blue text.
>
> Best regards,
> The authors

---

> > ### Comment · Reviewer_m3a9 · 2023-05-08
> > **Thank you for the update**
> >
> > I have read the updated manuscript, and the new Limitations section sufficiently incorporates my concerns.

---

### Author Response · Authors · 2023-04-22
**General Response to Reviewers / Additional Experiments**

We would like to thank all three reviewers for their detailed reviews and constructive feedback. We were glad that all reviewers concluded that the paper is well written, addresses an important and novel research question, and the study design and analytic methods are generally appropriate.

In response to the reviews, we significantly updated our study. We summarize the main updates to the paper below:

* We re-launched our user study through the platform Prolific. Our study now contains 1,330 responses from 190 participants, twice that of our original study.
* We updated the claims made throughout paper to reflect the insights obtained from the larger study we conducted.
* We added a thorough discussion of limitations of our work in the Discussion.

The revised paper marks the changes made in blue text.

---

### Decision · Action_Editors · 2023-05-18

**Recommendation:** Accept with minor revision

**Comment:**

The initial reviews were in broad agreement about the strengths and weaknesses of this paper. The reviewers found that while the proposed methodology was sound and the initial manuscript might make a good contribution to TMLR, it also had a few significant weaknesses.

The authors turned around and effectively provided a major revision within the discussion period. In a short amount of time they were able to address most of the weaknesses identified by reviewers. Notably, they were able to double the sample size of their study which allowed for stronger conclusions. They updated their analysis with this new data and updated the manuscript taking other reviewer comments into consideration. They also addressed other limitations of their work by adapting some of their claims (e.g., with respect to generality of the findings) and also adding a one-page discussion of limitations (Section 5.1).

The number of available features to the ML model (two) remains an important perceived limitation of this paper. This is clear from reviewers' comments and from our private discussion. I agree that this setting may not be that realistic (in this setting other features are likely available and so one would likely model them). I also do not know of the possible effects of increasing the number of features on the paper's findings apart from making the model more accurate. Having said that, I believe this point is less about correctness and more about significance of the paper and its results.

Given TMLR's evaluation criteria I find the paper should be accepted with minor revisions.


Minor revisions. I would like the authors to consider the following two changes:

- I found that some of the discussion you had with the reviewers regarding the above limitation (number of features provided to the model) was insightful and would be useful to add to the paper. In particular:
   - *Positioniong of your work with respect to the existing literature.* In your response you wrote
      > "The setting we consider is ML assisted decision making under limited information. In this context, one can consider scenarios where users have less than, more than, or the same features as the ML model. Each of these three scenarios has been studied by seminal works in the ML assisted decision making literature. For example, Poursabzi-Sangdeh et al. (2021) studied scenarios where users have the same features as the ML model and scenarios where users have more features than the ML model (in which case their ML model is also based on just two features). Kleinberg et al. (2017) consider scenarios where users have more features than the ML model."

       I did not find that this view was clearly exposed in the paper. Would it be possible to add it? If it is already there and I have missed it please kindly let me know.

    - *Discussing possible effects of using different features for the model and for humans.* In your reply to reviewers you also write
      > "As one of the first studies investigating how different types of posterior predictive distributions affect ML assisted decision making (and how such effects vary between domain/ML experts and non-experts), we did not want to introduce additional complications arising in the scenario where users have fewer features than the ML model. For instance, if the ML model includes some additional features (not available to the user) that are highly influential for a few apartments, model predictions may diverge heavily from experts’ understanding and create distrust in the model. We agree with the reviewer that exploring these additional scenarios would be interesting future directions. We will add a paragraph in the Discussion describing these additional scenarios one may consider."

        This seems to relate to some of the discussion in the second paragraph of Section 5.1. I would suggest adding some of the above to it (or somewhere else in the same section).

- Fixing the typos in the new text (in blue).

**Audience:**

The paper studies ML for human decision-making. In particular, the role and value of communicating model uncertainty estimate to human decision makers. This topic has clear applications for ML researchers working on uncertainty quantization and also for people working more directly at this intersection. While it is unclear how generalizable the findings in this paper are, it is clear that a number of individuals in TMLR's audience would be interested in them.

**Claims And Evidence:**

After reading the original manuscript, the reviewers made a series of suggestions to improve the evidence provided in the manuscript as well as to better delineate the scope of the resulting claims.

As a response, the authors launched an additional study during the review to gather more data. The additional data led to stronger conclusions. The authors also adapted the claims according to the reviewers' suggestions.

The claims are now supported by accurate, convincing, and clear evidence.